# Micro-costing analysis of a combination intervention for improved mental health and HIV risk behaviors among school-going adolescent girls in Uganda

Yesim Tozan[1]*, Joshua Kiyingi[2], Sooyoung Kim[3], Flavia Namuwonge[2], Florence Namuli[4], Vicent Ssentumbwe[2], Rashida Namirembe[4], Edwinnah Kasidi[4], Ozge Sensoy Bahar[2], Mary M. Mckay[2], Fred M. Ssewamala[2]

1 Department of Global and Environmental Health, School of Global Public Health, New York University, New York, New York, United States of America, 2 Brown School, Washington University in St. Louis, Saint Louis, Missouri, United States of America, 3 Department of Public Health Policy and Management, School of Global Public Health, New York University, New York, New York, United States of America, 4 International Center for Child Health and Development (ICHAD), Masaka, Uganda

* tozan@nyu.edu

**Data Availability Statement:** Data used in this analysis cannot be shared publicly due to privacy concerns and are available upon reasonable

## Abstract

Suubi4Her is a combination intervention that integrates a savings-led family-based economic empowerment intervention through youth development accounts with a family strengthening intervention delivered via multiple family groups. It aims to improve mental health and reduce HIV risk behaviors among school-going adolescent girls in Uganda. This micro-costing study was conducted as part of a three-armed randomized control trial between 2017–2022, involving 1,260 participants aged 14–17 years across 47 secondary schools. Adopting a provider perspective, we prospectively identified, measured, and computed the costs associated with all program activities. These costs were then aggregated and divided by the actual number of adolescent girls in each study arm to conservatively obtain the per-adolescent costs for each arm. The per-adolescent costs of economic empowerment intervention alone and in combination with the family strengthening intervention were US$476 and US$812, respectively. Personnel costs were the key cost driver due to the intensive supervision of intervention delivery and quality assurance efforts. This study is the first to estimate the economic costs of an evidence-based combination intervention targeting the multifaceted risk factors underlying HIV risk among adolescent girls in a low-resource setting. The per-adolescent cost of US$812 for the Suubi4Her intervention falls within the cost range reported for other family-based interventions (US$500-US$900); however, published comparisons are limited. Accurate and reliable cost estimates are key to assessing the feasibility, affordability, and economic value of interventions. There is a pressing need for more costing studies on evidence-based combination interventions, especially in low-resource settings (Trial registration: Clinical Trials NCT03307226; IRB approvals: Washington University in St. Louis (IRB #201703102), the Uganda Virus Research Institute (GC/127/17/07/619), and the Uganda National Council of Science and Technology (SS4406).

request. Data access requests can be directed to one of the following Associate Deans at Washington University's Brown School: Siomari Collazo-Colón, JD, Associate Dean for Administration, Hillman Hall, Room 254, Office Phone: 314-935-8675 (Email: scollazo@wustl.edu) or Byron Powell, PhD, Associate Dean for Research, Office Phone: 314-935-2817 (Email: bjpowell@wustl.edu). The research team is open to data sharing provided that the following conditions are met: 1) A formal research question is specified a priori; 2) The names, affiliations, and roles of any individuals who will access the shared data are disclosed; 3) The deliverables (e.g., manuscript, conference presentation) are specified a priori; 4) Proper credit and attribution for each deliverable, including authorship and co-authorship order, are established; 5) Requestors understand that the data cannot be further shared without the Principal Investigator's permission; and 6) Appropriate IRB approval is obtained for data use (or documentation that the IRB has determined the research is exempt). Requestors are expected to manage the conversion of electronic formats, although the research team may consider converting the data to a tab-delimited text format if feasible. These conditions were pre-specified in the study proposal, data sharing plan, and the consenting and assenting process. Given that this research involves a vulnerable population, the consent form stipulates that only de-identified individual-level data may be shared outside of the research team and only upon meeting the aforementioned conditions.

**Funding:** This work was supported by the National Institute of Mental Health (NIMH) [grant #: 1R01MH113486-01, PI: Fred M. Ssewamala, PhD]. The content is solely the responsibility of the authors and does not necessarily represent the official views of NIMH or the National Institutes of Health.

**Competing interests:** The authors have declared that no competing interests exist.

## Introduction

Despite notable progress in HIV prevention and treatment worldwide, HIV remains a significant public health challenge in sub-Saharan Africa (SSA) [1]. As of 2023, SSA is home to approximately 65% of people living with HIV (PLHIV), and accounts for about 50% of new HIV infections worldwide [2]. Adolescent girls and young women in SSA are particularly vulnerable to HIV due to a complex interplay of biological, behavioral, and socio-structural risk factors [3, 4]. Evidence shows that household financial instability, school dropout, and mental health challenges increase the risk of HIV infection among these groups as these factors found to be strongly associated with sexual risk-taking behaviors such as age-disparate or transactional sex and early marriage [5–10]. Against this backdrop, a multi-level response to address the complex and inter-related drivers of HIV risk among adolescent girls in SSA is essential to achieving the 95-95-95 targets set by the Joint United Nations Programme on HIV/AIDS goal by 2030 [1].

In the Central Region of Uganda, which has a disproportionately high HIV prevalence of 10.6% compared to the national average of 6.2%, [11] a randomized control trial "Suubi4Her" was implemented between 2017 and 2022 [12]. Aimed at addressing the multi-dimensional drivers of HIV risk among adolescent girls, the Suubi4Her intervention integrated an asset-based financial inclusion and economic empowerment model with a family strengthening intervention to reduce HIV risk behaviors. An evaluation of the short-term impact of the combination intervention showed that adolescent girls in the treatment arms experienced a significant reduction in the prevalence of depressive symptoms after 12 months of implementation [13]. The positive effects on mental health outcomes persisted at 24 months, however no significant effects on sexual risk-taking behaviors and attitudes were observed, potentially due to the young age of the target population [14].

Given the growing emphasis on informed healthcare spending, evidence on the effectiveness of interventions alone is insufficient to justify investments. In addition to evaluating outcomes, a comprehensive assessment of the economic costs associated with the delivery of interventions is critical for intervention planning, resource allocation decisions, and priority-setting in the context of ongoing resource limitations, growing budget constraints, and shifting health needs, particularly in low-resource settings [15]. However, there is limited information available on the costs of combination interventions [15, 16]. This study seeks to contribute to the currently limited economic evidence in this area by estimating the economic costs of the Suubi4Her intervention, an evidence-based combination intervention aimed at improving mental health and reducing HIV risk behaviors, with the goal of reducing new HIV infections among adolescent girls in Uganda.

## Methods

### Ethics statement

Written informed assent and consent were obtained from the adolescent participants and their caregivers, respectively. All study procedures were approved by Washington University in St. Louis (IRB #201703102), the Uganda Virus Research Institute (GC/127/17/07/619), and the Uganda National Council of Science and Technology (SS4406).

### Study population, setting and design

A three-arm cluster randomized-controlled trial was conducted in 47 secondary schools in Rakai, Kyotera, Masaka, Lwengo, and Kalungu districts in the Central Region of Uganda over a 5-year period (August 2017-July 2022) [11]. The trial is registered in the Clinical Trials

database (clinicaltrials.gov; registration number: NCT03307226) and described in detail else-where [12] and received IRB approvals IRB approvals from Washington University in St. Louis (IRB #201703102), the Uganda Virus Research Institute (UVRI) (GC/127/17/07/619), and the Uganda National Council of Science and Technology (UNCST) (SS4406).

A total of 1,260 school-going adolescent girls (aged 14–17 years at enrollment) were recruited between 3/23/2018 and 2/27/2019 and were randomized at the school level to one of the three study arms: 1) Control arm (UC, n = 16 schools, 408 girls) received usual care (UC) that was provided to all adolescents in the region; 2) Treatment arm 1 (UC+YDA, n = 16 schools, 471 girls) received UC and a savings-led FEE intervention through youth develop-ment accounts (YDAs); and 3) Treatment arm 2 (UC+YDA+MFG, n = 15 schools, 381 girls) received a family strengthening intervention delivered via multiple family groups (MFG) in the schools on top of what was provided in treatment arm 1.

Written informed assent and written consent were obtained from the adolescent partici-pants and their caregivers, respectively. All study procedures were approved by the Washing-ton University in St. Louis Review Board (IRB)–home institution of the PI–and by in-country local IRBs in Uganda: UVRI and UNCST—as mentioned earlier.

## Study interventions

Adolescents in the control arm received the Adolescent Sexual and Reproductive Health (ASRH) curriculum required for all secondary school students in Uganda, which was consid-ered UC in the study. In brief, the content of the ASRH curriculum is dispersed across various subjects and classes in secondary schools. Students learn about delaying sex, using contracep-tion, preventing forced sex and substance addiction, as well as gender equality and the impor-tance of postponing marriage. To ensure the consistent delivery of the curriculum over the trial period, induction seminars were conducted for the teachers in all the participating schools before program implementation.

Adolescents in treatment arm 1 received a savings-led economic empowerment interven-tion along with UC (UC+YDA). Specifically, participants received a YDA at a well-established and renown financial institution, co-held by adolescents and their caregivers. Family members and other relatives were encouraged to contribute to YDAs, and any savings deposited into the accounts were matched by the program at a 1:1 ratio, effectively doubling the contributions over the intervention period. In addition, four sessions of financial literacy training (FLT) were delivered by trained program staff to participants and their caregivers, aimed to provide them with basic financial knowledge. Only after the completion of FLT sessions, participants were given access to matching funds, which could be used to support either a family-based income generating business (up to 30%) or participants' education and skill development (at least 70%). All participants received financial incentives as a compensation of their time and effort.

Adolescents in treatment arm 2 received, in addition to the interventions in treatment arm 1, a family strengthening intervention via MFG, aimed at addressing common mental health challenges that affect adolescents (UC+YDA+MFG). A total of 16 sessions was delivered through task-shifting by 12 trained MFG facilitators, either community health workers (CHWs) or peer parents (PPs), under the supervision of program staff. Both CHWs and PPs were recruited from the communities (residing<1 mile from schools) and recommended by the schools' leadership to ensure trust and respect of the community members and to facilitate rapport and trusting relationships with participating families. The training of MFG facilitators consisted of a one-time 2-day session delivered by the program staff at the field office in Masaka. At the end of the training, all trainees completed a knowledge skills and attitude test.

Those who passed the test with a score of 85% or more received a certificate of completion and the manuals to deliver the MFG intervention. Due to disruptions caused by the COVID-19 pandemic, the delivery of MFG sessions continued until March 14, 2021. Each MFG session consisted of 12–20 families that included the adolescent girl-caregiver dyad plus any other family members, lasted 45–60 minutes, and involved activities such as roleplays, group discussions, and family activities. Every two weeks, MFG facilitators met with program staff to address concerns and report progress. Each MFG session was attended by two program staff to ensure consistency and fidelity in the delivery of the MFG intervention. All MFG facilitators and participants received financial incentives as a compensation of their time and effort. The costs of transportation incurred by participants to attend MFG sessions were also reimbursed.

## Cost data collection and analysis

To enhance the reliability and validity of the cost data, resource use in each study arm was captured prospectively over the trial period. Costs were then analyzed using an activity-based micro-costing approach from the program provider perspective. Micro-costing enumerates all resources used in intervention delivery and hence is considered the most reliable costing method [17]. The program provider perspective maintains the focus on costs pertinent to the implementation of an intervention and facilitates budgeting, financial planning, and technical efficiency analysis in the face of limited healthcare resources and competing interventions [18, 19].

To accurately capture resource use, we first identified all program-related activities in each study arm through a review of study protocols and administrative records and periodic interviews with key program staff. The shared activities across all three study arms included identification of schools and school visits, screening and recruitment of study participants, delivery of usual care, and stakeholder engagement and dissemination. We included the costs of school identification and visits and recruitment and screening processes as these costs may be incurred in real-life contexts depending on intervention's eligibility requirements, delivery method, and implementation setting. In each treatment arm, the activities involved in the delivery of the interventions were also identified. In treatment arm 1, these activities included the delivery of four FLT sessions that cover basic principles of financial management including income generation, use of financial institutions, saving and asset-building as well as opening of and contributing to YDAs. In treatment arm 2, YDA activities in treatment arm 1 were combined MFG-related activities, which included the training and supervision of MFG facilitators and the delivery of MFG sessions. The costs of the activities conducted purely for the purposes of research such as baseline and follow-up assessments of the study participants through interviews and surveys were excluded from the cost analysis because our objective was to estimate the resource requirements of the Suubi4her intervention in a non-research setting.

Next, we measured and valued all resources used for each program activity by study arm. Shared resources across program activities and study arms (i.e., personnel, program overheads, donated resources, and capital costs) were measured and valued as separate cost categories. Personnel included a program coordinator and other program staff who monitored the implementation of the interventions for quality assurance and worked as research assistants for research activities. Time costs of program staff were calculated based on average annual gross wage rates and time devoted to the Suubi4Her intervention over the trial period based on interviews with key program staff (S1 Text). Other recurring costs included monetary incentives provided to families, CHWs, PPs, and teachers (S1 Text), provision of educational materials, and program overheads, including utilities (electricity, water, communication), transportation to/from schools for implementation (fuel, taxi fare, car hire), security services and insurance, maintenance of equipment, vehicles, and facilities, office supplies and printing,

and other miscellaneous expenses. Expenditures and other pertinent data for the cost analysis were extracted from the financial and administrative records of the program. Where relevant, the costs related to a program activity were further broken down into and listed as cost items. For example, the delivery of the MFG sessions included participation incentives for families (time and transport), facilitation incentives for MFG facilitators (time and transport), community mobilization incentives for teachers (time and phone airtime) as well as the costs of training of the MFG facilitators and the donated classroom space.

We also considered and valued resources leveraged for the program activities to arrive at the economic costs of the interventions. Leveraged resources included the donated classroom space used for the screening and recruitment of study participants and delivery of the FLT and MFG sessions, and the donated time by school teachers and bank officials. Lastly, recurring costs were differentiated from capital costs, which typically include the costs of capital items with an expected useful life of more than one year (e.g., equipment, vehicles, and furniture). All capital costs were annualized using the replacement cost of each capital item, an appropriate useful life for each item, and an annual discount rate following the approach suggested by Walker and Kumaranayake [20]. This annualization method takes into account the opportunity cost of capital items, allowing us to estimate the economic cost of the capital in any year rather than its financial cost [20].

We then calculated the total cost per adolescent per study arm. To do this, all costs were first adjusted for inflation using the Ugandan Consumer Price Index (CPI) [21], discounted at an annual rate of 3% [22] to the start year of the intervention, and presented in 2018 US dollars [23]. The adjusted activity-based costs were then summed to calculate the total costs per study arm. Shared resources (program personnel costs, donated resources, overheads, and other capital costs) were apportioned to program (80%) activities versus research (20%) activities based on interviews with key program staff, and we only included the program costs in the analysis. By dividing the costs of the intervention activities in each study arm by the total costs of all intervention activities across the three arms we calculated ratios (UC, 17% vs. YDA, 35% vs. YDA+MFG, 48%) to apportion the costs of the shared resources to the three study arms. Total per-adolescent costs by study arm were calculated by dividing the total costs per study arm by the number of adolescent girls in each arm using the intent-to-treat (ITT) sample and the treatment-on-the-treated (TOT) sample. While the TOT sample includes persons who participated in an intervention, allowing a more conservative calculation of per-person costs, the cost analysis using the ITT sample yield per-person costs under the assumption that the intervention was taken up by all those who were offered. A summary of identified program activities, cost items, and measurement and calculation methods, is presented in Table 1. Our study follows the Consolidated Health Economic Evaluation Reporting Standards (CHEERS) and the CHEERS checklist is available in S1 Table.

## Results

Table 2 presents per-adolescent costs by cost category and study arm (UC, N = 420; UC+YDA, N = 420; and UC+YDA+MFG, N = 420) using the ITT sample. The per-adolescent cost for the study arms was estimated at $264 for the UC arm, $534 for the UC+YDA arm, and $737 for the UC+YDA+MFG arm. The incremental cost over the control arm was $270 for the UC +YDA arm and $472 for the UC+YDA+MFG arm. Personnel costs had the biggest share of the program costs across all study arms, followed by program overheads. In the treatment arms, the economic empowerment intervention activities (YDAs, FLT sessions, and IGA workshops) were key cost drivers, following personnel and program overheads. The implementation costs of the family strengthening intervention (MFG sessions) and the economic

**Table 1. Identification, quantification, and valuation of main cost categories.**

| Program activity/ Cost category | Cost items | Measurement and calculation methods |
|---|---|---|
| Activity-based costs | | |
| Identification of schools and school visits | Facilitation incentives for teachers (airtime for phones), provision of educational materials to teachers and students (teacher handbooks, student notebooks, pens, textbooks, and geometry sets), participation incentives for families | Divide the total cost incurred each year by number of participants in each study arm. |
| Screening and recruitment of study participants | Facilitation incentives for teachers (airtime for phones), participation incentives for families, interviewer compensation for screening of participants | Divide the total cost incurred each year by number of participants in each study arm. |
| Usual care | Provision of science textbooks to schools, and exercise books and pens to all participants in the study to improve on students learning | Divide the total cost incurred each year for textbooks, exercise books and pens by number of participants in each study arm. |
| Youth development accounts (YDA) | Account opening, initial deposit, matched contributions | Divide the cost of opening YDAs and matched contributions by number of participants in each treatment arm. |
| Delivery of financial literacy workshops | Reach the Youth-Uganda staff time to deliver FLT sessions, participation incentives for participants, and facilitation incentives for teachers | Divide the total cost incurred each year by number of participants in each treatment arm. |
| Delivery of income generating activities workshops | Reach the Youth-Uganda staff time to deliver IGA sessions, participation incentives for participants, and facilitation incentives for teachers | Divide the total cost incurred each year by number of participants in each treatment arm. |
| Training of MFG facilitators | Participation incentives for PPs and CHWs | Divide the total cost incurred each year by number of participants in treatment two arm. |
| Delivery of MFG sessions | Facilitation incentives for PPs and CHWs, facilitation incentives for teachers (airtime for phones) | Divide the total cost incurred each year by number of participants in treatment two arm. |
| Shared costs across activities | | |
| Personnel | Based on administrative and financial expenditure records of the study, time devoted to program activities (as opposed to research activities) in each study arm by the project coordinator and other program staff | Extract the number of hours each staff member devoted to program activities each year, multiply the total hours by average hourly salary rate of staff (estimated based on average annual gross wage rate), and calculate the total cost across all staff. Apportion the total cost incurred each year based on level of effort dedicated by staff to program activities (80%) (versus research activities, 20%) and to each study arm (17% control; 35% YDA only, and 48% YDA+MFG), and divide by number of participants in each study arm. |
| Donated resources | Time spent by teachers on mobilization of families for program-related activities, time spent by bank officials for YDA opening, classroom space used for program-related activities at participating schools | Based on an hourly time cost of teachers and bank officials and daily cost of classroom space, calculate the total cost incurred each year. Apportion the total cost to each study arm (17% control; 35% YDA only, and 48% YDA+MFG), and divide by number of participants in each study arm. |
| Program overheads | Utilities (water, electricity, communication), transportation (fuel, taxi fare, car hire), security services and insurance, maintenance (equipment, vehicles, and facilities), office supplies and printing, and other miscellaneous costs | Apportion the total cost incurred each year based on level of effort dedicated by staff to program activities (80%) (versus research activities, 20%) and to each study arm (17% control; 35% YDA only, and 48% YDA+MFG), and divide by number of participants in each study arm. |
| Stakeholder engagement and dissemination | Transport refund for teachers and stakeholders, and induction seminar-related and stakeholder meeting-related costs | Divide the total cost incurred each year by number of participants in each study arm. |
| Capital costs | Capital items with an expected useful life of more than one year, such as equipment, vehicles, and furniture | Calculate equivalent annual costs by annualizing all capital costs over the useful life of capital items (3 years for equipment, 5 years for furniture, and 10 years for vehicles), and apportion the total cost incurred each year based on level of effort dedicated by staff to program activities (80%) (versus research activities, 20%) and to each study arm (17% control; 35% YDA only, and 48% YDA+MFG), and divide by number of participants in each study arm. |

YDA = youth development account; IGA = income generating activity; FLT = financial literacy training; MFG = multiple family group; PP = peer parent; CHW = community health worker

**Table 2. Per-child costs of the economic empowerment and family strengthening intervention by study arm, using the intent-to-treat (ITT) sample (All costs are in 2018 Ugandan Shillings unless otherwise indicated).**

| Costs | Control arm (UC) | Treatment arm 1 (UC+YDA) | Treatment arm 2 (UC+YDA+MFG) |
|---|---|---|---|
| Personnel | 731,342 | 1,479,857 | 2,037,283 |
| *Salaries for program staff* | *731,342* | *1,479,857* | *2,037,283* |
| Identification of schools and school visits | 921 | 921 | 921 |
| *Facilitation incentives for teachers* | *921* | *921* | *921* |
| Screening and recruitment of study participants | 15,637 | 18,037 | 16,269 |
| *Participation incentives for families* | *12,468* | *14,513* | *12,946* |
| *Facilitation incentives for teachers* | *3,169* | *3,524* | *3,324* |
| Usual care | 93,966 | 93,966 | 93,966 |
| *Science textbooks* | *46,971* | *46,971* | *46,971* |
| *Exercise books* | *45,195* | *45,195* | *45,195* |
| *Pens* | *1,800* | *1,800* | *1,800* |
| Youth development accounts (YDA) | - | 86,900 | 78,539 |
| *Accounts opening and initial deposit* | *-* | *17,935* | *15,004* |
| *Matched contributions* | *-* | *68,964* | *63,535* |
| Delivery of FLT sessions | - | 21,143 | 23,400 |
| *Participation incentives for families* | *-* | *20,440* | *22,928* |
| *Facilitation incentives for teachers* | *-* | *703* | *472* |
| Delivery of IGA sessions | - | 5,466 | 4,612 |
| *Participation incentives for families* | *-* | *5,241* | *4,481* |
| *Facilitation incentives for teachers* | *-* | *225* | *131* |
| Training of MFG facilitators | - | - | 1,305 |
| *Participation incentives for CHWs and PPs* | *-* | *-* | *1,305* |
| Delivery of MFG sessions | - | - | 90,103 |
| *Participation incentives for families* | *-* | *-* | *49,597* |
| *Facilitation incentives for CHWs and PPs* | *-* | *-* | *39,057* |
| *Facilitation incentives for teachers* | *-* | *-* | *1,450* |
| Donated resources | 4,376 | 10,867 | 21,562 |
| *Screening and recruitment-classroom space* | *3,730* | *3,730* | *3,730* |
| *MFG-classroom space* | *-* | *-* | *10,806* |
| *FLT-classroom space* | *-* | *3,584* | *2,785* |
| *IGA-classroom space* | *-* | *1,369* | *1,172* |
| *Teachers' time* | *645* | *1,064* | *2,132* |
| *Bank officials' time* | *-* | *1,121* | *938* |
| Program overheads | 119,719 | 239,289 | 333,497 |
| *Utilities* | *4,254* | *5,649* | *11,850* |
| *Transportation* | *84,223* | *170,423* | *234,618* |
| *Communication* | *3,502* | *7,086* | *9,755* |
| *Maintenance* | *23,060* | *46,662* | *64,238* |
| *Materials and office supplies* | *4,680* | *9,470* | *13,037* |
| Stakeholder engagement and dissemination | 11,915 | 18,686 | 23,728 |
| *Stakeholder meetings* | *6,615* | *13,386* | *18,428* |
| *Induction seminars* | *5,300* | *5,300* | *5,300* |
| Capital costs | 7,051 | 14,269 | 19,643 |
| Total costs | 984,932 | 1,989,411 | 2,744,842 |

*(Continued)*

**Table 2.** (Continued)

| Costs | Control arm (UC) | Treatment arm 1 (UC+YDA) | Treatment arm 2 (UC+YDA+MFG) |
|---|---|---|---|
| Total costs (in 2018 USD) | 264 | 534 | 737 |

UC = usual care; YDA = youth development account; IGA = income generating activity; FLT = financial literacy training; MFG = multiple family group; PP = peer parent; CHW = community health worker

empowerment intervention were comparable, each accounting for about 3% of the total cost of the UC+YDA+MFG arm.

Table 3 presents the cost per-adolescent by cost category and study arm using the TOT sample. These cost estimates were derived based on the actual number of adolescents participated in the study per arm, making them higher and more conservative than those estimates by the ITT sample across all study arms. The UC arm had 408 adolescents, the UC+YDA arm 471 adolescents, and the UC+YDA+MFG arm 381 adolescents. Cost per-adolescent estimates based on the TOT sample were $272, $476 and $812 for the UC, UC+YDA, and UC+YDA +MFG arms, respectively. The incremental cost over the UC arm was $204 for the UC+YDA arm, and $540 for the UC+YDA+MFG arm.

## Discussion

In SSA, while access to HIV services has significantly improved since the early 2000s [1, 2], mental health services for children and adolescents remain severely limited, with substantial gaps in both policy and evidence [16, 24]. This is the first costing study of a combination intervention that combined an evidence-based family strengthening intervention together with a proven FEE intervention in the SSA context. Specifically, we expanded the current literature by estimating the economic costs of a combination intervention for adolescent girls in Uganda, a low resource setting. A key strength of this costing study is the prospective collection of resource use and cost data by trained staff alongside the Suubi4Her trial. We employed an activity-based micro-costing approach to improve the reliability and validity of cost estimates, a method particularly recommended for interventions without previous cost data [25]. The micro-costing approach, in contrast to the macro-costing approach, highlight the context-specific variations in the data, which is crucial for evaluating the feasibility and affordability of interventions within different implementation settings. Lastly, we closely followed the methodological recommendations of the Second Panel on Cost Effectiveness in Health and Medicine [18], carefully justifying the provider perspective based on the study's aim and adjusting the cost data appropriately, including currency conversion and inflation, discounting, and annuitization. Our approach aims to lay the ground for future studies evaluating the costs of combination interventions in other low-resource environments and the cost-effectiveness of this intervention which was shown to be effective in positively improving mental health outcomes of school-going adolescent girls in Uganda [26].

Our analysis showed that the cost per adolescent for delivering the FEE intervention through YDAs on top of UC was US$476, while the cost for the combined FEE and MFG intervention was US$812. Personnel costs represented the largest portion of the total costs, as the intervention activities demanded extensive supervision and quality assurance measures. The body of economic evidence on FEE interventions is currently limited. Previous studies in Uganda have costs for FEE interventions aimed at improving mental health outcomes for children and adolescents affected by HIV/AIDS, ranging between US$418 and US$426 per child [27]. Similarly, few studies have evaluated the costs of the MFG intervention. For example, a

**Table 3. Per-child costs of the economic empowerment and family strengthening intervention by study arm, using the treatment-on-the treated (TOT) sample (All costs are in 2018 Ugandan Shillings unless otherwise indicated).**

| Costs | Control arm (UC) | Treatment arm 1 (UC+YDA) | Treatment arm 2 (UC+YDA+MFG) |
|---|---|---|---|
| Personnel | 752,852 | 1,319,618 | 2,245,824 |
| *Salaries for program staff* | *752,852* | *1,319,618* | *2,245,824* |
| Identification of schools and school visits | 948 | 821 | 1,015 |
| *Facilitation incentives for teachers* | *948* | *821* | *1,015* |
| Screening and recruitment of study participants | 16,097 | 16,084 | 17,935 |
| *Participation incentives for families* | *12,834* | *12,942* | *14,271* |
| *Facilitation incentives for teachers* | *3,263* | *3,142* | *3,664* |
| Usual care | 96,730 | 83,792 | 103,585 |
| *Science textbooks* | *48,353* | *41,885* | *51,780* |
| *Exercise books* | *46,524* | *40,301* | *49,821* |
| *Pens* | *1,853* | *1,605* | *1,984* |
| Youth development accounts (YDA) | - | 77,490 | 86,578 |
| *Accounts opening and initial deposit* | *-* | *15,993* | *16,539* |
| *Matched contributions* | *-* | *61,497* | *70,039* |
| Delivery of FLT sessions | - | 18,854 | 25,795 |
| *Participation incentives for families* | *-* | *18,227* | *25,275* |
| *Facilitation incentives for teachers* | *-* | *627* | *520* |
| Delivery of IGA sessions | - | 4,874 | 5,084 |
| *Participation incentives for families* | *-* | *4,674* | *4,940* |
| *Facilitation incentives for teachers* | *-* | *201* | *144* |
| Training of MFG facilitators | - | - | 1,438 |
| *Participation incentives for CHWs and PPs* | *-* | *-* | *1,438* |
| Delivery of MFG sessions | - | - | 99,326 |
| *Participation incentives for families* | *-* | *-* | *54,674* |
| *Facilitation incentives for CHWs and PPs* | *-* | *-* | *43,054* |
| *Facilitation incentives for teachers* | *-* | *-* | *1,598* |
| Donated resources | 4,504 | 9,691 | 23,769 |
| *Screening and recruitment-classroom space* | *3,840* | *3,326* | *4,112* |
| *MFG-classroom space* | *-* | *-* | *11,912* |
| *FLT-classroom space* | *-* | *3,196* | *3,070* |
| *IGA-classroom space* | *-* | *1,221* | *1,292* |
| *Teachers' time* | *664* | *948* | *2,350* |
| *Bank officials' time* | *-* | *1,000* | *1,034* |
| Program overheads | 123,245 | 213,388 | 367,650 |
| *Utilities* | *4,380* | *5,039* | *13,067* |
| *Transportation* | *86,704* | *151,977* | *258,645* |
| *Communication* | *3,605* | *6,319* | *10,754* |
| *Maintenance* | *23,738* | *41,609* | *70,814* |
| *Materials and office supplies* | *4,817* | *8,444* | *14,371* |
| Stakeholder engagement and dissemination | 12,266 | 16,662 | 26,157 |
| *Stakeholder meetings* | *6,810* | *11,936* | *20,314* |
| *Induction seminars* | *5,456* | *4,726* | *5,843* |
| Capital costs | 7,259 | 12,724 | 21,654 |
| Total costs | 1,013,900 | 1,773,997 | 3,025,810 |

*(Continued)*

**Table 3.** (Continued)

| Costs | Control arm (UC) | Treatment arm 1 (UC+YDA) | Treatment arm 2 (UC+YDA+MFG) |
|---|---|---|---|
| Total costs (in 2018 USD) | 272 | 476 | 812 |

UC = usual care; YDA = youth development account; IGA = income generating activity; FLT = financial literacy training; MFG = multiple family group; PP = peer parent; CHW = community health worker

recent study in Uganda reported per-child costs of delivering the MFG intervention by PPs and CHWs at US$346 and US$328, respectively [15]. Pilot studies focused on improving parenting practices in Liberia and Thailand estimated per-beneficiary costs between US$650 and US$900 [28]. A South African study estimated the cost of a parenting program designed to prevent violence against adolescents at US$504 per family [29]. Compared to these studies, the FEE intervention's cost of US$476 was similar to existing findings, while the combination intervention's cost of US$812 per adolescent fell within the range reported in these studies. However, the limited number of costing studies underscores the need for further research in diverse implementation contexts and country settings.

While it remains uncertain if the levels of investments required for these interventions would be deemed acceptable to policy-makers in low- and middle-income countries (LMICs), there are compelling reasons to justify increased investment in adolescent health and wellbeing. One notable reason is the increasing proportion of children and adolescents within the overall LMIC population driven by improved survival rates and declining fertility rates [30] and the pressures the social and economic adversities place on transition to adulthood, particularly in resource-constrained settings [31]. Moreover, evidence increasingly supports the argument that investing in interventions that address multiple facets of adolescent health and wellbeing may yield significant long-term benefits. These benefits include improved health in adulthood and economic benefits for countries, with the highest return on investments in low-income and lower-middle countries [32, 33]. In light of this, there is a clear need for greater investment in interventions targeting child and adolescent health outcomes, as well as a pressing need for more research on the economic costs and cost-effectiveness of such interventions.

This study has several limitations. First, the time spent on intervention-related activities by program staff, teachers, and bank officials was estimated based on interviews with key personnel, which may have introduced bias. An alternative method is the use of daily logs to capture time-use data more accurately. However, this approach is time-intensive and can place a significant burden on program staff involved in the program activities. Second, this costing study was undertaken alongside a randomized controlled study. Costs associated with research activities, such as baseline and follow-up assessments conducted through interviews and surveys, were excluded from the analysis to focus on estimating the resource requirements for the interventions in a non-research setting. Nonetheless, some costs may still be ambiguous. For instance, monitoring, evaluation and quality assurance are crucial for the successful implementation of interventions in real-world settings. However, these costs are typically higher in research studies due to the more intensive nature of service delivery. Furthermore, the shared nature of certain resources such as program personnel, donated resources, program overheads, and capital items, complicates the task of allocating costs between program or research activities. Due to the lack of a precise allocation method, we applied conservative apportioning ratios based on the information provided by key program staff. Lastly, the interventions were delivered through task-shifting in school settings across specific districts of Uganda. The resource utilization and associated costs may vary based on the delivery approach and

geographical context, which limits the generalizability of our findings. This limitation is, however, balanced by the richness of the data micro-costing generates, providing a granular and comprehensive understanding of the inputs required for each intervention activity, which in turn allows for accurate and context-specific estimates. Further, the method highlights key cost drivers, enabling policymakers and program implementers to understand which factors contribute most to overall costs and where efficiency gains may be possible.

## Conclusions

Despite these limitations, this study leveraged prospectively data on resource use and costs alongside implementation, providing robust cost estimates for delivering a combination intervention to school-going adolescent girls in Uganda, a low-resource setting. As a result, the study provides policymakers and program implementers with essential information to assess the feasibility, affordability, and long-term sustainability of the intervention within their own context. Furthermore, the results lay the groundwork for conducting a cost-effectiveness analysis of the combination intervention in comparison to other proven interventions aimed at improving mental health outcomes and reducing HIV risk behaviors among adolescents in resource-limited settings, ultimately guiding resource allocation decisions in HIV prevention and mental health.

## Supporting information

**S1 Text. Additional description of the costing methods.**
(DOCX)

**S1 Table. CHEERS 2022 checklist: Micro-costing analysis of a combination intervention for improved mental health and HIV risk behaviors among school-going adolescent girls in Uganda.**
(DOCX)

## Acknowledgments

We thank Abel Mwebembezi of Reach the Youth, Uganda, Rev. Fr. Joseph Kato Bakulu of the Masaka Catholic Diocese, and the entire field team of the International Center for Child Health and Development (ICHAD) for their contributions to the study design and implementation. In addition, we are grateful to the financial institutions that agreed to work with the adolescent girls in opening savings accounts, and the extension workers who have committed time to train the adolescent girls in conducting income-generating activities. Our thanks also go to the Ugandan Government Ministry of Education and the 47 secondary schools that have agreed to participate in the Suubi4Her study.

## Author Contributions

**Conceptualization:** Yesim Tozan, Fred M. Ssewamala.

**Data curation:** Yesim Tozan, Joshua Kiyingi, Flavia Namuwonge, Florence Namuli, Vicent Ssentumbwe, Rashida Namirembe, Edwinnah Kasidi.

**Formal analysis:** Yesim Tozan, Joshua Kiyingi, Sooyoung Kim.

**Funding acquisition:** Mary M. Mckay, Fred M. Ssewamala.

**Investigation:** Yesim Tozan, Joshua Kiyingi, Sooyoung Kim, Flavia Namuwonge, Florence Namuli, Vicent Ssentumbwe, Rashida Namirembe, Edwinnah Kasidi, Ozge Sensoy Bahar, Mary M. Mckay, Fred M. Ssewamala.

**Methodology:** Yesim Tozan, Fred M. Ssewamala.

**Project administration:** Yesim Tozan, Ozge Sensoy Bahar, Mary M. Mckay, Fred M. Ssewamala.

**Resources:** Fred M. Ssewamala.

**Supervision:** Yesim Tozan, Fred M. Ssewamala.

**Validation:** Yesim Tozan, Joshua Kiyingi, Sooyoung Kim.

**Visualization:** Yesim Tozan, Joshua Kiyingi, Sooyoung Kim.

**Writing – original draft:** Yesim Tozan, Joshua Kiyingi, Sooyoung Kim.

**Writing – review & editing:** Flavia Namuwonge, Florence Namuli, Vicent Ssentumbwe, Rashida Namirembe, Edwinnah Kasidi, Ozge Sensoy Bahar, Mary M. Mckay, Fred M. Ssewamala.

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
