## [Decision Letter · Decision Letter 0]

26 Jul 2024

PMEN-D-24-00098

Micro-costing analysis of a combination intervention for improved mental health and HIV risk behaviors among school-going adolescent girls in Uganda

PLOS Mental Health

Dear Dr. Tozan,

Thank you for submitting your manuscript to PLOS Mental Health. We apologize for the delay in providing this feedback. It has been challenging to secure reviewers for your submission, which contributed to the delay. We have now received a detailed and thorough review from Reviewer 1, whose insights are crucial for enhancing the quality of your manuscript. Based on their review, and my assessment, we invite you to submit a revised version of the manuscript that addresses the points raised during the review process.

We look forward to receiving your revised manuscript.

Kind regards,

Andreas D Haas, PhD

Academic Editor

PLOS Mental Health

Journal Requirements:

1. Please include a complete copy of PLOS’ questionnaire on inclusivity in global research in your revised manuscript. Our policy for research in this area aims to improve transparency in the reporting of research performed outside of researchers’ own country or community. The policy applies to researchers who have travelled to a different country to conduct research, research with Indigenous populations or their lands, and research on cultural artefacts. The questionnaire can also be requested at the journal’s discretion for any other submissions, even if these conditions are not met.  Please find more information on the policy and a link to download a blank copy of the questionnaire here: https://journals.plos.org/plosmentalhealth/s/best-practices-in-research-reporting. Please upload a completed version of your questionnaire as Supporting Information when you resubmit your manuscript.

https://www.ajtmh.org/view/journals/tpmd/106/4/article-p1078.xml

https://bmcpublichealth.biomedcentral.com/articles/10.1186/s12889-023-15083-2

In your revision ensure you cite all your sources (including your own works), and quote or rephrase any duplicated text outside the methods section. Further consideration is dependent on these concerns being addressed.

Additional Editor Comments (if provided):

1) References in Introduction and Discussion: Please review and update the references throughout your manuscript. It is crucial that these references accurately support the claims you make. For instance, the data cited on the current HIV epidemic (Lines 56-58) rely on a reference from 2017. These figures are outdated.Please cite more recent data, for example the 2023 Spectrum estimates available at https://aidsinfo.unaids.org/.

2) First Sentence of the Discussion: The opening sentence of the discussion section needs reconsideration. While acknowledging that there are significant gaps in HIV care across sub-Saharan Africa, it is incorrect to state that HIV services are non-existent. Most countries in the region have established robust pediatric HIV programs. Please revise this statement to reflect a more accurate depiction of the HIV services landscape.

Reviewers' comments:

Reviewer's Responses to Questions

**Comments to the Author**

1. Does this manuscript meet PLOS Mental Health’s publication criteria? Is the manuscript technically sound, and do the data support the conclusions? The manuscript must describe methodologically and ethically rigorous research with conclusions that are appropriately drawn based on the data presented.

Reviewer #1: Yes

2. Has the statistical analysis been performed appropriately and rigorously?

Reviewer #1: Yes

3. Have the authors made all data underlying the findings in their manuscript fully available (please refer to the Data Availability Statement at the start of the manuscript PDF file)?

Reviewer #1: No

4. Is the manuscript presented in an intelligible fashion and written in standard English?

Reviewer #1: Yes

5. Review Comments to the Author

Reviewer #1: In this article, Tozan and colleagues present an economic analysis of the impact of the Suubi4Her combination intervention for adolescent girls in Uganda. The results are based on a randomized trial where eligible adolescents were divided into three groups: those receiving only education; those receiving also a financial support intervention; and those receiving, in addition, also a family strengthening intervention. The per-person costs for the two levels of intervention were estimated at about $500 and $800, which is comparable for other similar interventions.

I have the following comments and suggestions:

1) One major limitation is the lack of an estimate of the benefit. In the protocol at clinicaltrials.gov, there is no outcome related directly to mental health – is there a plan to quantify the benefit in terms of prevented mental health related disorders etc in the future? The protocol also mentions yearly cost-effectiveness calculations (which include also the estimation of the benefits, not only costs) – have these been performed in the past years? I understand that the estimation of benefits is a complex issue (and also will need more follow-up time), but you could in the Discussion give some rough ideas of the magnitude of the burden that could be potentially preventable. This could also put the cost estimates better into context ($800 per adolescent may sound a lot, but it could well be that the consequences of the mental health problems are even much higher).

2) Could you describe YDA still in a simpler way? Is it a requirement that family members / relatives also contribute, e.g. if the family pays $50 then the program will also pay $50 (is this what is meant by the 1:1 ratio)? How about those whose families cannot afford any payment? Is there any upper or lower limit for the amount that is given by the program? One could argue that this intervention helps most those who are already wealthy, i.e. whose families can afford big contributions.

3) With the payer perspective, are the costs limited to those paid externally, i.e. from the program? How about out-of-pocket costs to the families? Is it correct that the contributions of the families to YDA are not considered in the total costs?

4) The use of micro-costing approach can lead to very precise estimates - does this level of precision really reflect the level of confidence we have in the results? How well are the results generalizable to other similar settings? In my understanding, the micro-costing approach also ignores the potential synergies between different programs (would it not be more reliable to just rather at the end of the project compare the total costs in all settings)?

5) What is the rationale to use discounting in this project? I understand this is useful for future projections so that costs and benefits in the near future are given greater weight; but in the present analysis, the costs were anyway estimated retrospectively?

6. PLOS authors have the option to publish the peer review history of their article (what does this mean?). If published, this will include your full peer review and any attached files.

**Do you want your identity to be public for this peer review?** For information about this choice, including consent withdrawal, please see our Privacy Policy.

Reviewer #1: No

---

## [Editor Report · Decision Letter 1]

20 Nov 2024

Micro-costing analysis of a combination intervention for improved mental health and HIV risk behaviors among school-going adolescent girls in Uganda

PMEN-D-24-00098R1

Dear Dr Tozan,

We are pleased to inform you that your manuscript 'Micro-costing analysis of a combination intervention for improved mental health and HIV risk behaviors among school-going adolescent girls in Uganda' has been provisionally accepted for publication in PLOS Mental Health.

Best regards,

Andreas Haas

Academic Editor

PLOS Mental Health

In the introduction, the term "HIV acquisition" was replaced with "HIV infection." Please note that "HIV acquisition" is the preferred terminology, as some members of the community of people living with HIV perceive "HIV infection" as stigmatizing. If appropriate, please revert to "HIV acquisition." This change can be made during the proofing process unless instructed otherwise from journal staff.